# Serum Clusterin Concentration and Its Glycosylation Changes as Potential New Diagnostic Markers of SARS-CoV-2 Infection and Recovery Process

**DOI:** 10.3390/ijms25084198

**Published:** 2024-04-10

**Authors:** Katarzyna Sołkiewicz, Izabela Kokot, Monika Kacperczyk, Violetta Dymicka-Piekarska, Justyna Dorf, Ewa Maria Kratz

**Affiliations:** 1Department of Laboratory Diagnostics, Division of Laboratory Diagnostics, Faculty of Pharmacy, Wroclaw Medical University, Borowska Street 211a, 50-556 Wroclaw, Poland; izabela.kokot@umw.edu.pl (I.K.); m.b.kacperczyk@gmail.com (M.K.); 2Department of Clinical Laboratory Diagnostics, Medical University of Bialystok, Waszyngtona 15A St., 15-269 Bialystok, Poland; violetta.dymicka-piekarska@umb.edu.pl (V.D.-P.); justyna.dorf@umb.edu.pl (J.D.)

**Keywords:** COVID-19, serum clusterin glycosylation, lectin-ELISA

## Abstract

COVID-19 is an infectious disease caused by the SARS-CoV-2 virus. Glycoprotein clusterin (CLU) has many functions such as phagocyte recruitment, complement system inhibition, apoptosis inhibition, hormone and lipid transport, as well as in the immune response. The study aimed to assess the changes in CLU concentrations and the profile and degree of CLU glycosylation between patients with severe COVID-19, convalescents, and healthy subjects (control). The profile and degree of serum CLU N-glycosylation were analyzed using lectin-ELISA with specific lectins. CLU concentrations were significantly lower and relative reactivities of CLU glycans with SNA (*Sambucus nigra* agglutinin) were significantly higher in severe COVID-19 patients in comparison to convalescents and the control group. The relative reactivities of CLU glycans with MAA (*Maackia amurensis* agglutinin), together with relative reactivity with LCA (*Lens culinaris* agglutinin), were also significantly higher in patients with severe COVID-19 than in convalescents and the control group, but they also significantly differed between convalescents and control. The development of acute inflammation in the course of severe COVID-19 is associated with a decrease in CLU concentration, accompanied by an increase in the expression of α2,3-linked sialic acid, and core fucose. Both of these parameters can be included as useful glycomarkers differentiating patients with severe COVID-19 from convalescents and the control group, as well as convalescents and healthy subjects.

## 1. Introduction

Severe acute respiratory syndrome coronavirus 2 (SARS-CoV-2) has been identified as the cause of the COVID-19 (coronavirus disease 2019) pandemic, which officially was announced by the WHO in March 2020 [1]. The SARS-CoV-2 infection in humans could be asymptomatic or accompanied by flu-like symptoms such as, e.g., fever, sore throat, cough, back pain, and loss of smell [2]. In severe cases of COVID-19, pneumonia, systemic inflammation, acute respiratory distress syndrome, or tissue or cardiac damage could occur, resulting in the patient’s death [3]. The group at increased risk includes the elderly and immunocompromised subjects, as well as those suffering from co-existing diseases such as diabetes, obesity, hypertension, and others [4]. Clusterin (CLU), also known as apolipoprotein J (ApoJ), is a 75–80 kDa heterodimeric, highly glycosylated extracellular glycoprotein. It is present in almost all body tissues and fluids and plays a multidirectional biological role in the human body [5,6], including inhibition of the complement system and cell apoptosis, as well as participation in the immune response [7,8]. Clusterin can be found in many body fluids, including blood plasma, and may exist in two forms: the highly glycosylated secretory form (sCLU) and its non-glycosylated intracellular nuclear form (nCLU), which still is not well characterized [5,6]. During maturation, the sCLU-precursor is N-glycosylated and cleaved into two disulfide-linked glycosylated subunits designated as α (34–36 kDa) and β (36–39 kDa). The deglycosylated, protein part of the CLU α chain is 24 kDa, and for β, it is 28 kDa, suggesting that the carbohydrate part of each subunit is approximately 30% of its molecular mass [9]. CLU contains six N-glycosylation sites: Asn-86, Asn-103, Asn-145, Asn-291, Asn-354, and Asn-374. Seven types of CLU N-glycans are known, which include mono- or di-sialylated bi-antennary oligosaccharide structures with and without fucose, sialylated tri-antennary glycans with and without fucose, and sialylated tetra-antennary sugar structures. The most dominant glycoform has two antennas, which are di-sialylated and not fucosylated [10]. The biological role of this glycoprotein has been proven in diseases with accompanying acute inflammation, which is related to the properties of CLU such as regulating the formation of the membrane attack complex (MAC) of complement components C5b, C6, and C7 and modulation of pro-inflammatory processes, which include NF-κB signaling and several cytokines, inter alia tumor necrosis factor α (TNF-α) and interleukin-2 (IL-2) [11]. Secretory clusterin has chaperone properties, and its activity depends on the degree of glycosylation [12,13,14]. CLU glycosylation is a part of the proper biosynthesis of CLU, as well as being necessary for the formation of soluble high-molecular-weight complexes of clusterin with misfolded proteins which are removed by endocytosis and lysosomal degradation in an ATP-independent manner. This process can occur both inside and outside cells, and is extremely important in maintaining homeostasis [15]. One of the most prominent changes occurring in the glycan structure of CLU seems to be associated with sialylation and fucosylation [16,17,18], which caused them to become the subject of our research. The present study aimed to assess the profile and degree of clusterin fucosylation and sialylation in blood sera of patients with COVID-19, convalescents, and healthy subjects, and it was carried out using a modified semi-quantitative lectin-ELISA. We were interested in whether the profile and degree of CLU glycosylation differ between the examined groups of participants, and together with CLU concentration determinations may be used as potential additional diagnostic markers associated with SARS-CoV-2 infection.

## 2. Results

The CLU concentration values in examined sera and relative reactivities of CLU glycans with lectins used are presented in Table 1 as mean values, and standard deviations (SD) as well as an interquartile range of received values. The median and distribution of CLU concentration values and CLU glycans relative reactivities with lectins tested, measured for severe COVID-19 patients, convalescents, and the control group, are presented in Figure 1. The Spearman rank correlations between analyzed parameters are shown in Table 2.

The concentration values of CLU and relative reactivities of CLU glycans with lectins used are presented as mean ± SD and Q1–Q3 in Table 1, and median values together with distributions in all three examined groups are shown in Figure 1. CLU concentrations and the relative reactivities of CLU glycans were significantly lower in severe COVID-19 patients than in the convalescents and control group (*p* = 0.007441 and *p* = 0.0001, respectively). The expression in CLU glycans of SNA-reactive α2,6-linked sialic acid was significantly higher in severe COVID-19 patients in comparison to convalescents (*p* = 0.00015) and control group (*p* = 0.02512). The relative reactivities of CLU glycans with MAA specific to α2,3-linked sialic acid were significantly lower in the control group than those observed for the severe COVID-19 patients and convalescents (*p* = 0.000000 and *p* = 0.005094, respectively). Additionally, the relative reactivities of CLU glycans with MAA were significantly lower in convalescents than in severe COVID-19 patients (*p* = 0.000001). The expression of LCA-reactive core fucose in CLU glycans was significantly lower in convalescents and control group (*p* = 0.00339 and *p* = 0.000000, respectively) than for severe COVID-19 patients. The relative reactivities of CLU glycans with fucose-specific LTA were significantly higher in severe COVID-19 patients than those observed in the control group (*p* = 0.000996). The results of the Spearman rank test used for correlation analysis between examined parameters are shown in Table 2. A high negative correlation between CLU concentrations and the relative reactivities of CLU glycans with LTA was observed. The average and weak negative correlations were also observed between CLU concentrations and the relative reactivities of CLU glycans with MAA, LCA, and UEA. The average and weak positive correlations were also observed between relative reactivities of CLU glycans with sialo- and fucose-specific lectins but high positive correlations exist between MAA and LCA relative reactivities.

The results of the ROC (receiver operating characteristics) curve analysis are shown in Figure 2 and Table 3. The results of the ROC curve analysis performed for severe COVID-19 patients and healthy subjects (control) significantly differ in values of three parameters: CLU concentrations, CLU glycan reactivity with MAA, and CLU glycan reactivity with LCA. For differentiation between severe COVID-19 patients and convalescents, three parameters were also identified: CLU glycan reactivities with SNA, MAA, and LCA. The convalescents and control group were significantly differentiated by CLU glycan reactivities with MAA and LCA. For the determination of cut-off points for each relative reactivity of CLU glycans with lectins, the Youden index method was used. The verification of the clinical value of the laboratory test was based on the value of the area under the curve (AUC) and can be defined as zero (0–0.5), limited (0.5–0.7), moderate (0.7–0.9), and high > 0.9 [19]. The results of the ROC curve analysis were shown only for parameters for which AUC was higher than 0.700 (Table 3, Figure 2).

## 3. Discussion

Although more than 80% of SARS-CoV-2-infected persons have no or mild symptoms and recover quickly from infection, there is a group of subjects with critical symptoms that require rapid diagnosis and effective treatment. In the present study, we investigated the glycosylation profile and degree of blood serum CLU in patients hospitalized in the intensive care unit due to severe COVID-19, convalescents, and healthy subjects. To the best of our knowledge, this is the first report presenting the results of blood serum CLU glycosylation analysis in the context of the course of COVID-19. The most serious complications of COVID-19 include i.a. sepsis-like inflammation, and respiratory or cardiovascular complications. In a typical viral infection, the immune system triggers an immediate inflammatory response to limit the spread of infection. In severe cases of SARS-CoV-2 infection, inflammation progresses to hyperinflammation, causing tissue damage which leads to organ failure [20]. Viral infection and developing inflammation cause changes in hepatic synthesis and blood plasma concentrations of acute-phase proteins. From a point of view of the present study, it is important to note that Begue et al. [21], who quantified 14 blood plasma apolipoproteins in patients with COVID-19 and in healthy subjects, observed significantly reduced concentrations in 9 of them, including CLU, in the course of SARS-CoV-2 infection (CLU concentration values were within the range of 20–230 µg/mL) in comparison to CLU levels in healthy subjects (CLU concentration values were within the range of 60–350 µg/mL). Even though our CLU concentration values were within a much narrower concentration range, both in the group of severe COVID-19 patients and in healthy persons, the results of our investigations follow the findings of Begue et al. [21] as we also observed that serum CLU concentrations were significantly lower in patients with severe SARS-CoV-2 infection in comparison to convalescents and the control group of healthy subjects. The lack of significant differences in CLU concentrations between convalescents and the control group may indicate that during the recovery process, the level of this glycoprotein increases to the values observed in healthy subjects who have never had COVID-19, which makes this parameter a promising marker of full recovery after SARS-CoV-2 infection.

In their review article, Al-Kuraishy et al. [22] concluded that reducing high-density lipoproteins (HDL) in COVID-19 is connected to the disease severity and poor clinical outcomes, suggesting that high HDL serum levels could benefit COVID-19. SARS-CoV-2 binds HDL, and this complex is attached to the co-localized receptors, facilitating viral entry. Therefore, SARS-CoV-2 infection may induce the development of dysfunctional HDL through different mechanisms, including the induction of inflammatory and oxidative stress with the activation of inflammatory signaling pathways. On the other hand, the induction of dysfunctional HDL induces the activation of inflammatory signaling pathways and oxidative stress, increasing COVID-19 severity. In the recent past, Tanaka et al. [23] reported that acute inflammation in septic shock patients is accompanied by a decrease in the concentration of blood plasma HDL, which indicates the existence of analogous associations between the low concentration of HDL and the occurrence of acute inflammation, as it is in the case of severe SARS-CoV-2 infection. As CLU is a normal component of blood plasma HDL particles [9,24], the observed in the present study decrease in CLU concentrations, together with decreased HDL levels (Appendix A), in the course of severe SARS-CoV-2 infection and the presence of positive correlations between CLU and HDL levels, confirms the findings of the above-mentioned authors.

Our study aimed to check whether the profile and degree of serum CLU glycosylation differ between patients with severe COVID-19, convalescents, and healthy subjects who did not suffer from this disease and did not have antibodies against the SARS-CoV-2 virus, and such differences were observed by us. We demonstrated a significantly higher expression of α2,6-linked SA in serum CLU glycans in the course of severe SARS-CoV-2 infection when compared with convalescents and the control group of healthy subjects. Moreover, the significantly lower expression of α2,3-linked sialic acid was observed in the control group in comparison to the other two examined groups, and this parameter additionally allowed us to differentiate patients with severe COVID-19 from convalescents. This may indicate that the increase in SA expression observed during severe SARS-CoV-2 infection, mainly this α2,6-linked, after recovery, returns to the level observed in healthy subjects. Taking into account the fact that the increased expression of SA on glycoprotein glycans is manifested during the development of acute inflammation in the course of various diseases, including those of viral origin, and the reduction in its expression is associated with recovery, such a relationship observed between severe COVID-19 patients and healthy persons who have previously been infected with SARS-CoV-2 makes this parameter an additional promising marker of the recovery process of severe COVID-19 patients.

Depriving CLU glycans of sialic acid may result in a change in its molecular conformation, which may affect the stability of the molecule, antigenic expression, or recognition of its receptors [25]. In the present study, we observed the presence of moderate negative correlations between serum CLU concentrations and relative reactivities of its glycans with MAA specific to α2,3-linked sialic acid. Moreover, we also observed strong negative correlations between CLU concentrations and relative reactivities of CLU glycans with fucose-specific LTA (Table 2), which may suggest that in the case of severe COVID-19, low concentrations of serum CLU are accompanied by a high expression of α2,3-linked SA and α1,3-linked antennary fucose typical for sialo-Lewis^x^ oligosaccharide structures. The present study also showed that the expression of LCA-reactive core fucose in CLU glycans was significantly higher in COVID-19 patients when compared to convalescents and healthy subjects. The observed negative correlations between CLU concentrations and relative reactivities of CLU glycans with LCA additionally confirm that together with a decreased CLU concentration increase fucosylation of this glycoprotein, which seems to be characteristic of blood serum CLU in the course of severe SARS-CoV-2 infection. As an increased expression of fucose on glycoprotein glycans is known as another marker of inflammatory condition, also here, in the course of severe COVID-19, CLU glycans highly decorated by fucose of Lewis^x^ oligosaccharide structures as well as core fucose, together with increased SA expression, seems to be typical for this viral disease.

To check the usefulness of the obtained results in differentiating examined groups, an ROC curve analysis was performed. Based on the obtained results, we observed that of the six analyzed parameters, only two of them, namely the expression of α2,3-linked SA and core fucose, differentiate severe COVID-19 patients and convalescents from the control group as well as convalescents from SARS-CoV-2-infected patients (AUC > 0.700, Table 3), confirming our above-mentioned observations. CLU concentration, with a cut-off point of 25.43 µg/mL, has a moderate clinical value (AUC 0.704) and may be a useful parameter distinguishing patients with severe COVID-19 from healthy subjects. The relative reactivity of CLU glycans with SNA has also a moderate clinical value (AUC 0.711) and differentiates severe COVID-19 patients and convalescents.

Importantly, two of the five tested glycomarkers are particularly interesting, expression in the CLU glycans of 2,3-linked sialic acid and core fucose, because their expression differs significantly not only between the group of severe COVID-19 patients vs. convalescents and healthy subjects, but we also observed the presence of significant differences in the values of these parameters between healthy persons and convalescents, with the simultaneous lack of significant differences between these groups in CLU concentrations. This is probably because the increased degree of sialylation and fucosylation during COVID-19 normalizes over a longer period than in the case of CLU concentration, and 3–4 weeks after the onset of the disease is too short a time for this process to be fully normalized. The so-called phenomenon ‘silent inflammation’ may occur here, which is still manifested by increased expression of SA and fucose compared to the state of full health not preceded by a viral infection, despite the return of CLU concentrations to the levels observed in healthy persons who have never had SARS-CoV-2 infection. This may indicate that the healing process, despite the disappearance of the symptoms of the disease, has not ended at all molecular levels.

The cascade reactions that follow complement activation can be initiated in at least three different ways. One of them is the lectin pathway (LP). The LP allows the activation of the complement through the recognition of glycan structures on the surface of i.a. viruses by mannose-binding lectin (MBL). The complement cascade terminates with the formation of the MAC [26]. Fox and Parks [27] reported that CLU prevents efficient MAC insertion into host cell membranes and inhibits complement-mediated lysis of host cells as well as prevents C5b-7 insertion into virions or host cell membranes. Complement evasion is critical for pathogens to survive this potent host immune response. Still, the open question is how coronaviruses evade complement-mediated lysis of infected cells [27]. Because both sialic acid and fucose of glycoprotein glycans participate in, among others, glycoprotein–cell and cell–cell interactions modulating the course of many disease processes, and because one of the biological roles of CLU is the inhibition of the complement activation, such types of interactions should also be taken into account in the context of the spread of SARS-CoV-2 infection. This hypothesis seems very likely, as the significant negative correlations between CLU concentrations and the degree of its α2,3 sialylation and fucosylation were especially exposed in severe COVID-19 patients. Moreover, in the course of severe SARS-CoV-2 infection vs. convalescents and healthy persons, decreased CLU concentrations accompanied by increased glycosylation, as observed in the case of α2,3-linked SA and core fucose expression, were noticed. The above insights may indicate a special role of glycosylation of this glycoprotein in the course of severe COVID-19. In future studies, it would certainly be worth checking whether CLU can be recruited to cells infected with SARS-CoV-2 in a manner dependent on its glycosylation and whether the glycosylation of this glycoprotein plays a role in the mechanism of CLU binding to coronavirus-infected cells and/or the dissemination of this mechanism among circulating SARS-CoV-2 viruses.

## 4. Materials and Methods

### 4.1. Patient Samples

The study was conducted following the Helsinki-II declaration and the protocol was approved by the Bioethics Committee of the Medical University of Bialystok (Permission No. APK.002.26.2021, APK-002.171.2023) and Bioethics Human Research Committee of the Wroclaw Medical University (Permission No. KB-36/2023, KB-89/2023). Informed written consent was obtained from each study participant. The serum samples were collected from patients who were admitted to the Emergency Department of the University Clinical Hospital in Bialystok between January and November 2021 with active SARS-CoV-2 infection, with diagnosis confirmed using a polymerase chain reaction (PCR) test (n = 87; age 56–75). The group of COVID-19 patients included subjects at 3rd and 4th stages based on Modified Early Warning Score (MEWS) classification who required intensive treatment because of pneumonia, with or without acute respiratory distress syndrome (ARDS), and with or without multiple organ dysfunction syndrome (MODS) [28]. None of the SARS-CoV-2-infected patients were in the 1st or 2nd stage of the disease (exclusion criteria). The Polish Society of Epidemiology and Infectious Diseases recommended the MEWS score for identification of the stage of COVID-19, which relies on the following parameters: body temperature, blood pressure, heart and respiratory rate, and neurological symptoms [28]. Based on the above parameters, four stages of COVID-19 progression were described: (1) asymptomatic and mildly symptomatic infection, (2) symptomatic infection with pneumonia, without symptoms of ARDS, (3) symptomatic infection with pneumonia and symptoms of ARDS, and (4) symptomatic infection with MODS (Appendix A). The project included patients with severe COVID-19 who were conscious and able to make decisions about participation in the study. The characteristics of severe COVID-19 patients are presented in Table 4. The group of convalescents and healthy subjects of the control group had to meet the criteria described below.

The convalescents group was composed of 50 subjects (age 28–75, 19 males/31 females), with positive anti-SARS-CoV-2 IgG antibodies in blood, who had suffered from SARS-CoV-2 infection in the last 3–4 weeks before recruitment to the study, did not take any medications, and for whom the course of the disease was mild (increased temperature, loss of taste and smell, headache, fatigue, and muscle pain). None of the patients required hospitalization, and they were SARS-CoV-2-negative at the time of collecting blood serum for the present study. Convalescents who qualified for the study did not receive any anti-inflammatory drugs at the time of blood collection for the study. The control group consisted of 65 healthy participants (age 30–74, 27 males/38 females) who did not suffer from SARS-CoV-2, and whose blood was free from specific IgG antibodies against this virus. Both convalescents and healthy subjects from the control group with comorbidities were excluded from participation in the study.

### 4.2. Biological Material

From all study participants, 5.5 mL of venous blood was collected in test tubes without anticoagulants (S-Monovette, SARSTEDT, Nümbrecht, Germany). Next, within 30 min, blood was centrifuged for 20 min at 1000× *g* to obtain serum, which was further stored at −75 °C at the Department of Clinical Laboratory Diagnostics Medical University of Bialystok. Blood sera transported from the Medical University of Bialystok were stored at −86 °C at the Wroclaw Medical University Biobank until the start of the research. Prior to performing the assay, all serum samples were gradually defrosted and mixed using a Vortex.

### 4.3. Clusterin Concentration

Blood serum clusterin concentrations were determined using commercially available immuno-enzymatic ELISA test (Human Clusterin Elisa Kit from Invitrogen, ThermoFisher Scientific, catalog No. EHCLU; Frederick, MA, USA), according to the manufacturer’s instruction, without any modifications. In short, anti-human CLU antibody has been pre-coated in the wells of the supplied microplate, and in the first step, 100 µL of standards (for the standard curve) and samples to the appropriate wells was added and bound to the immobilized (capture) antibody (incubation for 2.5 h at room temperature with gentle shaking). Next, the plate was washed with 4-times-diluted wash buffer, and after washing, 100 µL of prepared biotin conjugate was added to each well (incubation for 1 h at room temperature with gentle shaking). After washing, 100 µL of prepared Streptavidin-HRP solution was added to each well, and the plate was incubated for 45 min at room temperature with gentle shaking. In the next step, after washing, 100 µL of TMB substrate was added to each well, and after incubation for 30 min at room temperature in the dark, 50 µL of Stop Solution was added to each well. After gentle mixing, the absorbance of colored complex formed, and CLU concentrations were read at 450 nm within 30 min after adding the Stop Solution using Mindray-96A reader (Mindray, Shenzhen, China). A standard curve and blanks were included in each ELISA test. When the samples were diluted for the analysis, the final CLU concentrations were calculated as concentrations obtained from ELISA plate reader multiplied by the appropriate dilution factor. The coefficient of variations CV% (for determination of intra-assay and inter-assay precision) was defined by the manufacturer. The intra-assay CV% for this test was <10% and inter-assay CV% was defined as <12%. The determined concentrations of clusterin were used to calculate constant amounts of CLU (50 ng CLU/100 µL), which were applied to the wells of the ELISA plate to analyze the glycosylation of this glycoprotein.

### 4.4. Lectin-ELISA

The wells of ELISA plate (Nunc MaxiSorp, Thermo Fisher Scientific, Glostrup, Denmark) were coated by goat anti-human clusterin polyclonal antibodies (Invitrogen, Thermo Fisher Scientific, catalog no. PA1-26903; Rockford, IL, USA) diluted in a ratio of 1:10,000 for SNA, and 1:5000 for MAA, LCA, LTA, and UEA in 10 mM TBS, pH = 8.5. After 2 h incubation at 37 °C, the plate was washed three times using the same buffer. Free binding sites were blocked by 10 mM TBS, 0.1% Tween20, 1% BSA, pH = 7.5 (blocking buffer); next, the plate was incubated for 2 h at 37 °C and then stored at 4 °C overnight. For LCA, due to the high absorbance of blanks in the preliminary experiments, oxidation of oligosaccharides of anti-human clusterin polyclonal antibodies, which coated ELISA plate, was performed. Sodium meta-periodate solution (100 mM NaIO_4_, 100 mM NaHCO_3_, pH = 8.1) was added, and after a 90 min incubation at room temperature in the dark, the plate was washed with 10 mM TBS, pH = 7.5. In the next step, free binding sites of ELISA plate wells were blocked by 10 mM TBS, 0.1% Tween20, 1% BSA (blocking buffer, pH = 7.5). After 2 h of incubation at 37 °C, plates were stored at 4 °C overnight. In the lectin-ELISA procedure with LTA and UEA, the step of oligosaccharide oxidation of anti-human clusterin polyclonal antibodies was unnecessary and therefore omitted. The serum samples were diluted in 10 mM TBS 0.1% Tween20 to obtain 50 ng CLU/100 µL, applied to each well of the ELISA plate, and incubated at 37 °C for two hours with gentle shaking. All samples were analyzed in duplicate to minimize imprecision. For each lectin-ELISA experiment, two pairs of blanks and internal control sera with known lectin reactivity were included. Blanks contained all reagents, but instead of patient samples, 10 mM TBS, 0.1% Tween20, pH = 7.5 (washing buffer) was used. After each step of lectin-ELISA, the wells were washed using a washing buffer. In the next step, biotinylated lectins (Vector Laboratories Inc., Burlingame, CA, USA) detecting α2,6- and α2,3-linked sialic acid (SNA and MAA, respectively) diluted in a ratio of 1:10,000 and 1:5000, respectively, and fucose-specific lectins LCA, LTA, and UEA in a dilution of 1:5000 were used (the exact specificity of lectins are presented in Table 5). Then, the plates were incubated for one hour at 37 °C with gentle shaking. To detect the clusterin–lectin complexes, ExtrAvidin alkaline phosphatase labeled (Sigma-Aldrich, catalog no. E2636; Saint Louis, MO, USA), diluted in a ratio of 1:10,000 in the washing buffer, was used. Next, plates were incubated for one hour at 37 °C, and then, the color reaction with disodium para-nitrophenyl phosphate was induced. The absorbances were measured with Mindray MR-96A Microplate Reader (Mindray Bio-Medical Electronics, Shenzen, China) at 405 nm with a reference filter 630 nm. The relative reactivities of CLU glycans with specific lectins were expressed in absorbance units (AU), after subtracting the absorbances of the blank samples. The lectin-ELISA method used in the present study for CLU glycosylation analysis was based on the method described by us previously [17,18].

### 4.5. Statistical Analysis

Statistical analysis was performed using the Statistica 13.3PL software package (StatSoft Inc., Tulsa, OK, USA). Shapiro–Wilk’s test was used to analyze the normality of all examined parameter distributions. Due to the lack of confirmation of a normal distribution for results obtained, nonparametric tests were used. Values of CLU concentrations and relative reactivities of CLU glycans with specific lectins were presented as mean ± SD (SD—standard deviation) and interquartile range (Q1–Q3)—see Table 1 as well as the graphs showing median values (Figure 1) The ANOVA test was used to compare CLU concentration values and relative reactivities of CLU glycans with lectins between analyzed groups of participants. The correlations between examined parameters with a 95% confidence interval were estimated according to the Spearman rank test. A two-tailed *p*-value of less than 0.05 was considered significant. The utility of lectins’ relative reactivities with CLU glycans and CLU concentrations for COVID-19 diagnostics was analyzed using receiver operating characteristic (ROC) curves. Based on the AUC (area under the curve) measurement, the clinical value of laboratory tests can be defined as zero (0–0.5), limited (0.5–0.7), moderate (0.7–0.9), and high (>0.9) [19]. The Youden index method was used for the determination of cut-off points. The *p*-values (probability values) less than 0.05 were considered significant.

## 5. Conclusions

To our knowledge, this is the first report analyzing changes in CLU glycosylation during severe SARS-CoV-2 infection. For the first time, a cut-off point was also proposed for the value of serum CLU concentrations, a parameter of moderate clinical value that differs significantly between severe COVID-19 patients vs. convalescents and healthy subjects. The expression in the CLU glycans of 2,3-linked sialic acid and core fucose is particularly interesting as potential diagnostic glycomarkers differentiating severe COVID-19 patients from convalescents and healthy persons, as well as healthy subjects and convalescents. The lectin-ELISA method does not provide information on the sugar composition of individual CLU glycans, as in the case of, e.g., GC-MS; however, the aim of our study was not to determine in detail the structure of carbohydrate units expressed on CLU present in the sera of severe COVID-19 patients, convalescents, and healthy subjects, but the analysis of changes in the relative content of glycotopes available for specific lectins. Interactions with lectins in vitro most likely reflect the interactions of glycans with lectins that occur in vivo between glycoconjugates of glycoproteins containing glycans conformationally accessible to their specific endo- and exogenous receptors, and have additionally enabled us to deepen our knowledge of the molecular mechanisms of these reactions. However, it should be remembered that most of the lectins used do not have absolute specificity and therefore may bind to similar carbohydrate structures with different affinities. Additionally, due to the semi-quantitative nature of the methods used, to obtain more complete information on the topic of CLU glycosylation changes in the course of severe COVID-19 in comparison to convalescents and healthy subjects, we see the need to continue research using more advanced quantitative methods. We believe that our research will contribute to a better understanding of the mechanisms related to the spread of this virus via the interaction of SARS-CoV-2-infected cells with CLU, setting directions for further research.

## Figures and Tables

**Figure 1 ijms-25-04198-f001:**
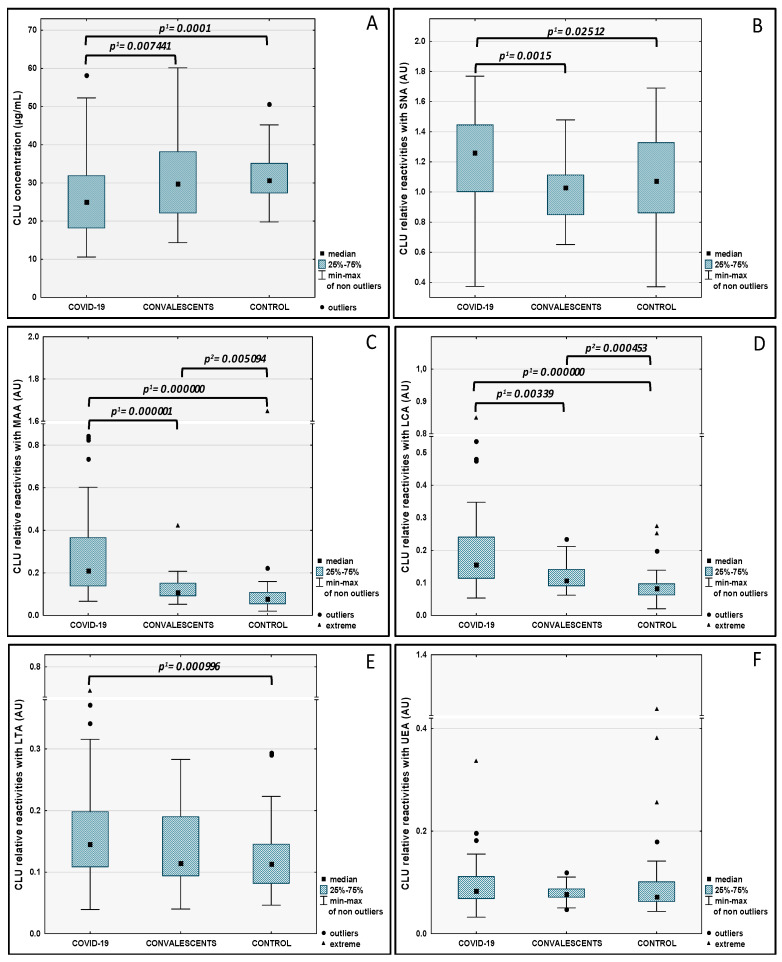
Clusterin concentrations (**A**) and relative reactivities of serum clusterin glycans with specific lectins (**B**–**F**). Significant differences between healthy subjects (control) vs. ^1^ patients with severe COVID-19 and ^2^ convalescents. CLU glycan relative reactivities with lectins SNA (*Sambucus nigra* agglutinin), MAA (*Maackia amurensis* agglutinin), LCA (*Lens culinaris* agglutinin), LTA (*Lotus tetragonolobus* agglutinin), and UEA (*Ulex europaeus* agglutinin) were expressed in absorbance units (AU). The exact specificity of the lectins used can be found in Section 4.4. A two-tailed *p*-value (probability value) of less than 0.05 was considered significant.

**Figure 2 ijms-25-04198-f002:**
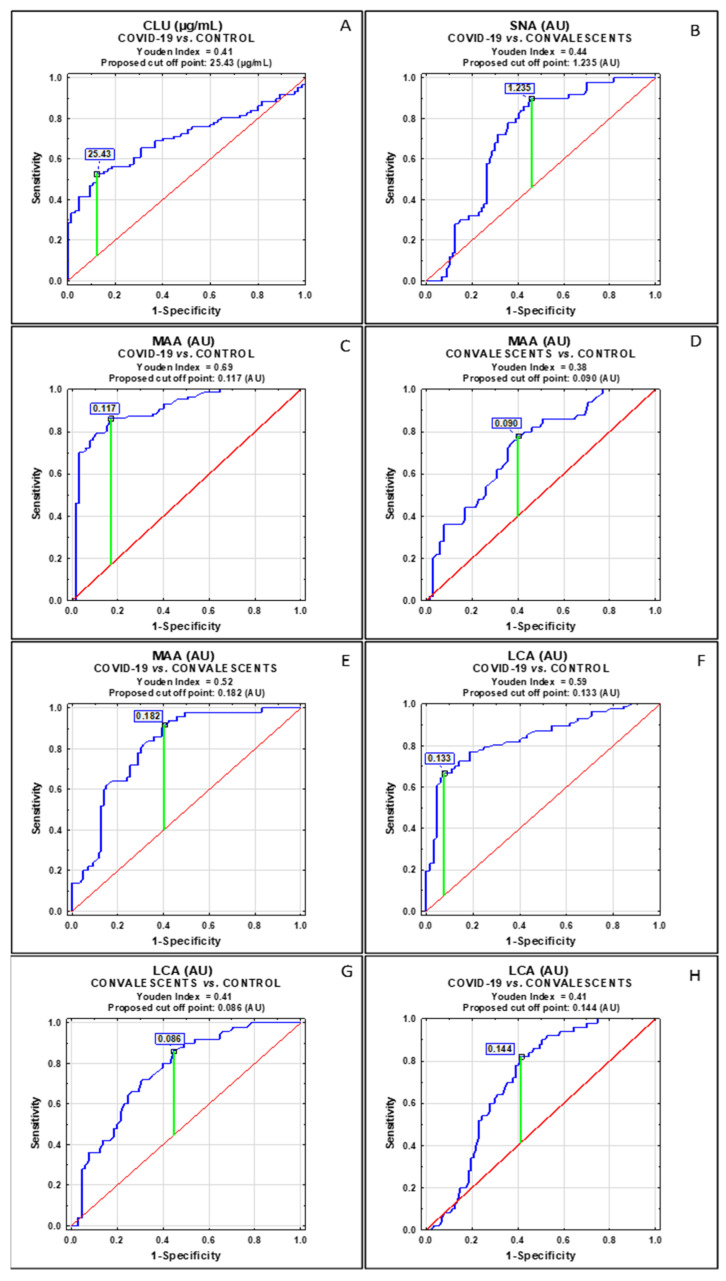
ROC curve analysis for clusterin concentrations (**A**) and relative reactivities of CLU glycans with lectins (**B**–**H**). Only the parameters for which the values of area under the curve (AUC) were ≥0.704 (moderate and high clinical value) were presented. For the determination of cut-off points, the Youden index method was used. The reference line is marked in red, the receiver operating characteristics for analyzed parameter in blue, and the cut-off point in green. CLU—blood serum clusterin concentrations; SNA—CLU glycan relative reactivity with *Sambucus nigra* agglutinin; MAA—CLU glycan relative reactivity with *Maackia amurensis* agglutinin; LCA—CLU glycan relative reactivity with *Lens culinaris* agglutinin. Significant differences were accepted for a *p*-value (probability value) of less than 0.05. The exact specificity of the lectins used can be found in Section 4.4.

**Table 1 ijms-25-04198-t001:** Concentrations of serum clusterin and relative reactivities of clusterin glycans with specific lectins.

	Parameters	CLU Concentration (µg/mL)	Relative Reactivity with Lectins (AU)				
Study Groups			SNA	MAA	LCA	LTA	UEA
COVID-19 n = 87		26.26 ± 10.08 (18.20–31.88)	1.203 ± 0.302 (1.003–1.446)	0.273 ± 0.183 (0.139–0.366)	0.190 ± 0.121 (0.114–0.241)	0.166 ± 0.103 (0.109–0.198)	0.092 ± 0.043 (0.067–0.112)
CONVALESCENTS n = 50		32.36 ± 11.85 (22.13–38.15) *p* ^1^ *=* 0.007441	1.018 ± 0.203 (0.849–1.113) *p* ^1^ *=* 0.00015	0.122 ± 0.060 (0.092–0.151) *p* ^1^ *=* 0.000001	0.119 ± 0.038 (0.091–0.141) *p* ^1^ *= 0.00339*	0.138 ± 0.066 (0.094–0.190)	0.078 ± 0.015 (0.071–0.087)
CONTROL n = 65		31.76 ± 6.20 (27.37–35.12) *p* ^1^ *=* 0.0001	1.090 ± 0.282 (0.862–1.328) *p* ^1^ *=* 0.02512	0.106 ± 0.198 (0.055–0.108) *p* ^1^ *=* 0.000000 *p* ^2^ *=* 0.005094	0.088 ± 0.044 (0.063–0.098) *p* ^1^ = 0.000000 *p* ^2^ = 0.000453	0.120 ± 0.055 (0.082–0.145) *p* ^1^ = 0.000996	0.106 ± 0.158 (0.063–0.101)

The concentrations of CLU in sera and the relative reactivities of CLU glycans with lectins are presented as mean values ± SD (standard deviation) with interquartile range Q1–Q3 (in brackets). Significant differences are shown for control (healthy subjects) versus ^1^ patients with severe COVID-19 and ^2^ convalescents. AU—absorbance units; CLU—clusterin; LCA—*Lens culinaris* agglutinin; LTA—*Lotus tetragonolobus* agglutinin; MAA—*Maackia amurensis* agglutinin; n—number of patients; SNA—*Sambucus nigra* agglutinin; UEA—*Ulex europaeus* agglutinin. The exact specificity of the lectins used can be found in Section 4.4. Significant differences were accepted for a *p*-value (probability value) of less than 0.05.

**Table 2 ijms-25-04198-t002:** The correlations between CLU concentrations and relative reactivities of clusterin glycans with lectins.

Correlations between Examined Parameters	Spearman Rank Coefficient (r)	*p*-Value
CLU and MAA	r = −0.365527	*p* = 0.000000
CLU and LCA	r = −0.461278	*p* = 0.000000
CLU and LTA	r = −0.687168	*p* = 0.000000
CLU and UEA	r = −0.376033	*p* = 0.000000
SNA and MAA	r = 0.354233	*p* = 0.000000
SNA and UEA	r = 0.466746	*p* = 0.000000
SNA and LCA	r = 0.349245	*p* = 0.000000
MAA and LTA	r = 0.244900	*p* = 0.000444
MAA and UEA	r = 0.372314	*p* = 0.000000
MAA and LCA	r = 0.588511	*p* = 0.000000
LTA and UEA	r = 0.301986	*p* = 0.000013
LTA and LCA	r = 0.413250	*p* = 0.000000
UEA and LCA	r = 0.309854	*p* = 0.000007

The table presents the significant correlations between analyzed parameters. CLU—blood serum clusterin concentrations; SNA—CLU glycan relative reactivity with *Sambucus nigra* agglutinin; MAA—CLU glycan relative reactivity with *Maackia amurensis* agglutinin; LCA—CLU glycan relative reactivity with *Lens culinaris* agglutinin; LTA—CLU glycan relative reactivity with *Lotus tetragonolobus* agglutinin; UEA—CLU glycan relative reactivity with *Ulex europaeus* agglutinin. The exact specificity of the lectins used can be found in Section 4.4. Significant differences were accepted for a *p*-value (probability value) of less than 0.05.

**Table 3 ijms-25-04198-t003:** Summary of the results of ROC curves analysis of clusterin concentrations and relative reactivities of CLU glycans with specific lectins.

Parameter	Study Groups		AUC	AUC with 95% Confidence Interval	Cut-off Point	Sensitivity	Specificity	*p*-Value
CLU	COVID-19	vs. CONTROL	0.704	0.622–0.786	25.43	0.529	0.877	0.0000
	CONVALESCENTS		0.471	0.354–0.588	43.85	0.220	0.969	0.6271
	COVID-19 vs. CONVALESCENTS		0.646	0.551–0.741	18.33	0.264	0.960	0.0025
SNA	COVID-19	vs. CONTROL	0.623	0.534–0.712	1.230	0.575	0.662	0.0071
	CONVALESCENTS		0.579	0.475–0.684	1.235	0.900	0.338	0.1369
	COVID-19 vs. CONVALESCENTS		0.711	0.625–0.797	1.235	0.900	0.540	0.0000
MAA	COVID-19	vs. CONTROL	0.906	0.858–0.955	0.117	0.862	0.831	0.0000
	CONVALESCENTS		0.726	0.635–0.818	0.090	0.780	0.600	0.0000
	COVID-19 vs. CONVALESCENTS		0.812	0.740–0.883	0.182	0.920	0.598	0.0000
LCA	COVID-19	vs. CONTROL	0.840	0.777–0.904	0.133	0.667	0.923	0.0000
	CONVALESCENTS		0.763	0.677–0.849	0.086	0.860	0.554	0.0000
	COVID-19 vs. CONVALESCENTS		0.709	0.624–0.795	0.144	0.820	0.552	0.0000
LTA	COVID-19	vs. CONTROL	0.676	0.590–0.761	0.135	0.598	0.692	0.0001
	CONVALESCENTS		0.573	0.465–0.680	0.158	0.380	0.831	0.1840
	COVID-19 vs. CONVALESCENTS		0.587	0.485–0.689	0.114	0.520	0.724	0.0954
UEA	COVID-19	vs. CONTROL	0.427	0.335–0.520	0.045	0.080	0.969	0.1236
	CONVALESCENTS		0.489	0.381–0.597	0.095	0.900	0.323	0.8430
	COVID-19 vs. CONVALESCENTS		0.593	0.499–0.686	0.095	0.900	0.414	0.0515

The analysis was performed for all three groups of study participants. The values of area under the curve (AUC) ≥ 0.704 and the corresponding rows are marked in grey. For the determination of cut-off points, the Youden index method was used. The clinical value of a laboratory test, based on AUC value, can be defined as zero (0–0.5), limited (0.5–0.7), moderate (0.7–0.9), and high (>0.9). CLU—blood serum clusterin concentrations; SNA—CLU glycan relative reactivity with *Sambucus nigra* agglutinin; MAA—CLU glycan relative reactivity with *Maackia amurensis* agglutinin; LCA—CLU glycan relative reactivity with *Lens culinaris* agglutinin; LTA—CLU glycan relative reactivity with *Lotus tetragonolobus* agglutinin; UEA—CLU glycan relative reactivity with *Ulex europaeus* agglutinin. The exact specificity of the lectins used can be found in Section 4.4. Significant differences were accepted for a *p*-value (probability value) of less than 0.05.

**Table 4 ijms-25-04198-t004:** Characteristics of patients with severe COVID-19.

COVID-19 Patients, n = 87											
Parameters											
Sex n (%)		Age		Length of Hospital Stay		Comorbidities		Comorbidities		Symptoms	
Male n (%)	Female n (%)	Years	n (%)	Days	n (%)	Absent n (%)	Present n (%)	Type	n (%)	Type	n (%)
35 (40%)	52 (60%)	≤55	21 (24%)	≤10	41 (47%)	36 (41%)	51 (59%)	Hypertension	42 (48%)	Cough	
										absent	9 (10%)
										present	78 (90%)
		56–75	28 (32%)	11–20	32 (37%)			Coronary artery disease	33 (38%)	Fever	
										absent	13 (15%)
										present	74 (85%)
		≥76	38 (44%)	≥21	14 (16%)			Diabetes mellitus	28 (32%)	Dyspnea	
										absent	14 (16%)
										present	73 (84%)
								Obesity	11 (13%)	Respiratory	
										Failure	
										absent	7 (8%)
										present	80 (92%)
								Other (hematological, cancer)	4 (5%)		

n—number of patients.

**Table 5 ijms-25-04198-t005:** Specificity of the lectins used in the study.

Lectin	Specificity for Sugar Moiety of N-glycan
*Sambucus nigra* agglutinin (SNA)	recognize sialic acid α2,6-linked [29]
*Maackia amurensis* agglutinin (MAA)	recognize sialic acid α2,3-linked [29]
*Lens culinaris* agglutinin (LCA)	recognize sequences containing fucosylated tri-mannose N-glycan core sites [30]
*Lotus tetragonolobus* agglutinin (LTA)	recognize antennary fucose α1,3-linked to GlcNAc [31]
*Ulex europaeus* agglutinin (UEA)	recognize antennary fucose α1,2-linked to Gal and α1,3-linked to GlcNAc [32]

Gal—galactose; GlcNAc—N-acetylglucosamine. Based on Sołkiewicz et al. [33].

## Data Availability

The data supporting this study’s findings are available from the corresponding author upon reasonable request.

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
