# Peer review of "Serum Clusterin Concentration and Its Glycosylation Changes as Potential New Diagnostic Markers of SARS-CoV-2 Infection and Recovery Process"

_ijms, 2024, doi:10.3390/ijms25084198_

Round 1

Reviewer 1 Report

Comments and Suggestions for Authors

The manuscript by SoÅ‚kiewicz et al., describes the analysis of the glycoprotein Clusterin concentration and its glycosylation profile in 3 groups of blood sera samples from i) severe COVID-19 patients, ii) convalescents, and iii) healthy subjects. The concentration of Clusterin was determined by ELISA test, as indicated in the Materials and Methods section. However, this Reviewer consider that a more explicit description of CLU quantification is necessary for the readers both in the Introduction and, expecially, in the Results sections. In contrary, the glycoprofile analysis via lectin-ELISA test is well described all along the main text. The authors found differences in both the secretory CLU concentration and in the relative glycan content that, according to this Reviewer, has an important implication in immune outcome, via host lectin mediated immune response, and as potential diagnostic markers. Thus, overall this Reviewer considers that the work is of high significance for a broad scientific community and deserves publication. Neverthless, few points need to be further analyzed. 

In line 104, Results Section, the Authors claim that the sCLU concentrations were significantly lower in severe COVID-19 patients than in the convalescents and control groups. Although this result is in line with that described by Begue et al., ref 21, the results from this work do not really show a “significant” difference in sCLU concentrations among the 3 groups, which instead look very similar with the interquartile range Q1-Q3 indicating no significant differences among them. Interestingly, in the Discussion and Conclusion sections, the Authors admit that their results do not really support such a defference as found by Begue et al. Thus, terms such as “significant” should be used carfully. 

Regarding reactivity against the panell of lectins, similar inconsistency can be found. Specifically, from line 107, the authors discuss the results of lectin ELISA assay as “significant” in all the cases: SNA “significantly higher in severe COVID-19 patients in comparison to convalescents”; MAA line 110: “significantly lower in the control group”; line 114 LCA “significantly lower in convalescents and control group”; line 117, LTA “significantly higher in severe COVID-19 patients”. However, table 1 and figure 1 show significat differences mainly for MAA and LCA but not clearly for SNA, LTA and UEA. For those lectins, in fact, the difference is not so clear and in case of MAA and UEA (control group) the SD is higher than the average value. 

Thus, as general observation, while in the Result and Discussion sections the authors indicate as significant all the differences observed, in contrary, in the conclusion section (line 259-261) they claim that only two parameters (alpha 2,3 and core fucose) “differentiate severe COVID-19 patients and convalescents from the control group as well as convalescents from SARS-CoV-2 infected patients”. This Reviewer strongly support this conclusion, while considers the discussion of the results for SNA, LTA and UEA an overstatment.

I suggest to be more consistent with results analysis and discussion. 

In section 4.4, the lectin-ELISA procedure for sCLU glycosylation analysis is not described but referred to references 17 and 18. I consider important to report also here the description of the analysis, at least with few lines.

Along the manuscript the ELISA test was described as semi-quantitative. However, a quantitative analysis requires competition experiments with glycans. Please specify how and why the ELISA is defined as semi-quantitative.

As main limitation of the conclusion this reviewer envisages: i) lectin-ELISA is the only technique used to analyse the glycoprofile. A second technique should be used to corroborate the results; ii) lectin promisquity may lead to unclear results. Please describe these limitations. 

Author Response

Reviewer 1

Open Review

(x) I would not like to sign my review report
( ) I would like to sign my review report Quality of English Language

( ) I am not qualified to assess the quality of English in this paper
( ) English very difficult to understand/incomprehensible
( ) Extensive editing of English language required
( ) Moderate editing of English language required
( ) Minor editing of English language required
(x) English language fine. No issues detected

Yes

Can be improved

Must be improved

Not applicable

Does the introduction provide sufficient background and include all relevant references?

(x)

( )

( )

( )

Are all the cited references relevant to the research?

(x)

( )

( )

( )

Is the research design appropriate?

( )

(x)

( )

( )

Are the methods adequately described?

( )

(x)

( )

( )

Are the results clearly presented?

( )

(x)

( )

( )

Are the conclusions supported by the results?

(x)

( )

( )

( )

Comments and Suggestions for Authors

The manuscript by SoÅ‚kiewicz et al., describes the analysis of the glycoprotein Clusterin concentration and its glycosylation profile in 3 groups of blood sera samples from i) severe COVID-19 patients, ii) convalescents, and iii) healthy subjects. The concentration of Clusterin was determined by ELISA test, as indicated in the Materials and Methods section. However, this Reviewer consider that a more explicit description of CLU quantification is necessary for the readers both in the Introduction and, expecially, in the Results sections. In contrary, the glycoprofile analysis via lectin-ELISA test is well described all along the main text. The authors found differences in both the secretory CLU concentration and in the relative glycan content that, according to this Reviewer, has an important implication in immune outcome, via host lectin mediated immune response, and as potential diagnostic markers. Thus, overall this Reviewer considers that the work is of high significance for a broad scientific community and deserves publication. Neverthless, few points need to be further analyzed. 

Answer: We would like to thank the Reviewer for the time devoted to preparing the review of our manuscript and for all the constructive comments that allowed us to improve the quality of our article. If we understand the Reviewer's comment regarding the description of CLU quantification correctly, the description of the method we used to determine CLU concentrations should be included in the Materials and Methods section (subsection 4.3) and it has been supplemented there. In our opinion, this is the best place to post this type of information. All corrections made in the manuscript were highlighted in green.

  • In line 104, Results Section, the Authors claim that the sCLU concentrations were significantly lower in severe COVID-19 patients than in the convalescents and control groups. Although this result is in line with that described by Begue et al., ref 21, the results from this work do not really show a “significant” difference in sCLU concentrations among the 3 groups, which instead look very similar with the interquartile range Q1-Q3 indicating no significant differences among them. Interestingly, in the Discussion and Conclusion sections, the Authors admit that their results do not really support such a defference as found by Begue et al. Thus, terms such as “significant” should be used carfully. 

Answer: Bergue and coworkers reported a significant difference (p < 0.0001, marked as **** in Figure 2) in serum clusterin concentrations between two groups of participants - a group of soldiers suffering from COVID-19, without differentiation between severity of disease, and a control group of healthy subjects. Unlike the study by Begue et al., in our study three groups of participants were analyzed - patients with severe COVID-19, convalescents, and a control group of healthy persons. Similarly to results obtained by Bergue et al., we showed significantly (according to statistical analysis rules, the differences between analyzed groups are significant for p-value <0.05) lower CLU concentrations in the group of severe COVID-19 patients when compared to the other two study groups (p = 0.0074 and p = 0.0001, respectively) as it was mentioned in the Results section. In our conclusions, however, we focused only on the results of ROC curves analysis, which indicated parameters of high clinical value, characterized by both high sensitivity and specificity, that may be used in routine diagnostics to differentiate groups of patients with severe COVID-19, convalescents, and healthy subjects.

  • Regarding reactivity against the panell of lectins, similar inconsistency can be found. Specifically, from line 107, the authors discuss the results of lectin ELISA assay as “significant” in all the cases: SNA “significantly higher in severe COVID-19 patients in comparison to convalescents”; MAA line 110: “significantly lower in the control group”; line 114 LCA “significantly lower in convalescents and control group”; line 117, LTA “significantly higher in severe COVID-19 patients”. However, table 1 and figure 1 show significat differences mainly for MAA and LCA but not clearly for SNA, LTA and UEA. For those lectins, in fact, the difference is not so clear and in case of MAA and UEA (control group) the SD is higher than the average value. 

Answer: When describing the obtained results, all significant differences between individual groups in the relative reactivities of CLU glycans with lectins were discussed, assuming that p-values of less than 0.05 are significant. This is consistent with the information given in subsection 4.5 Statistical Analysis. When presenting our results, we do not eliminate outliers and extreme values (Figure 1) that result in high values of standard deviations (SD) as in the case of MAA- and UEA-reactivity in control group. However, it should be underlined that the conclusions were formulated based on the results of ROC curves analysis.

  • I suggest to be more consistent with results analysis and discussion.

Answer:  Thank you for this comment, however, it is too enigmatic to us and gives us no chance to change the formula of results analysis in the Discussion section. We have made every effort to provide a structured and logical description of the significant changes in sCLU glycosylation between analyzed groups of participants that we have demonstrated in our study, compared to recent results obtained by other investigators. Each of the authors of scientific studies has his style of describing and discussing the results, which, of course, does not always have to fully meet the readers' expectations, however, to change anything, specific indications are needed, which is unclear because authors are usually 'attached' to their texts and the way of discussing the obtained results and from this perspective, it is difficult for them to see what might not be fully understandable to others.

  • In section 4.4, the lectin-ELISA procedure for sCLU glycosylation analysis is not described but referred to references 17 and 18. I consider important to report also here the description of the analysis, at least with few lines.

Answer: The description of sCLU glycosylation analysis using biotinylated lectins presented in the manuscript is detailed and does not omit any steps (subsection 4.4, in green). References No 17 and 18 refer to our previous studies of CLU glycosylation where this methodology was described for the first time in the context of CLU glycosylation analysis.

  • Along the manuscript the ELISA test was described as semi-quantitative. However, a quantitative analysis requires competition experiments with glycans. Please specify how and why the ELISA is defined as semi-quantitative.

Answer: Due to the lack of availability of commercial glycan standards (branched oligosaccharides), the lectin-ELISA tests we use to assess the glycosylation profile and degree are based on semi-quantitative determinations (glycans relative reactivities with lectins expressed in absorbance units). Based on the relative reactivities of glycoprotein glycans with lectins used, measured in individual samples, we can confirm or exclude the existence of glycans in glycoprotein tested as well as the differences in expression of examined glycans between tested groups. This method does not provide information on the sugar composition of individual CLU glycans, as in the case of e.g. GC-MS, however, the aim of our study was not to determine in detail the structure of carbohydrate units expressed on CLU present in the sera of severe COVID-19 patients, convalescents, and healthy subjects, but analysis of changes in the relative content of glycotopes available for specific lectins. Interactions with lectins most likely reflect similar types of interactions that may occur in vivo between glycoconjugates of glycoproteins containing glycans conformationally accessible to their specific endo- and exogenous receptors, hence the use of the lectin-based methods, including lectin-ELISA, also provides interesting information. The above information was supplemented in the Conclusions section.

  • As main limitation of the conclusion this reviewer envisages: i) lectin-ELISA is the only technique used to analyse the glycoprofile. A second technique should be used to corroborate the results; ii) lectin promisquity may lead to unclear results. Please describe these limitations. 

Answer: The limitations regarding the use of the lectin-ELISA as well as the need to continue the analysis of blood serum CLU glycosylation in study participants using quantitative techniques are described in the Conclusions section.

Reviewer 2 Report

Comments and Suggestions for Authors
  1. The main question  undertaken by this study is whether is clusterin concentration determinations could be used as potential diagnostic markers associated with SARS-CoV-2 infection, which in my opinion is important and valuable in the field.
  2. The study addressed biochemistry changes (Clusterin), as a marker to infectious diseases and diagnostic method fields.  
  3. I think there the study will give attention to a new trend in evaluating the incidence and progress of  SARS-CoV-2 infection.
  4. However, specific improvements should be considered regarding the general language, methodology, further controls regarding comparing the results with tradition methods such PCR or serology the should be considered.
  5. Abstract and conclusion may also need to be improved, as the objective and aim are not included in the abstract, while further recommendations should be included in the conclusion.
  6. I think references are appropriate and updated.
  7. I think the data of the results are well presented in tables and figures with satisfied quality.
Comments on the Quality of English Language

Needs a little bit improvement

Author Response

Reviewer 2

Open Review

( ) I would not like to sign my review report
(x) I would like to sign my review report Quality of English Language

( ) I am not qualified to assess the quality of English in this paper
( ) English very difficult to understand/incomprehensible
( ) Extensive editing of English language required
( ) Moderate editing of English language required
(x) Minor editing of English language required
( ) English language fine. No issues detected

Yes

Can be improved

Must be improved

Not applicable

Does the introduction provide sufficient background and include all relevant references?

( )

(x)

( )

( )

Are all the cited references relevant to the research?

(x)

( )

( )

( )

Is the research design appropriate?

(x)

( )

( )

( )

Are the methods adequately described?

(x)

( )

( )

( )

Are the results clearly presented?

(x)

( )

( )

( )

Are the conclusions supported by the results?

(x)

( )

( )

( )

Comments and Suggestions for Authors

  1. The main question  undertaken by this study is whether is clusterin concentration determinations could be used as potential diagnostic markers associated with SARS-CoV-2 infection, which in my opinion is important and valuable in the field.

Answer: We would like to thank the Reviewer for this opinion. We would also like to thank you for taking the time to review our manuscript and for all your valuable comments.

  1. The study addressed biochemistry changes (Clusterin), as a marker to infectious diseases and diagnostic method fields.  

Answer: Thank you for this comment.

  1. I think there the study will give attention to a new trend in evaluating the incidence and progress of  SARS-CoV-2 infection.

Answer: We would like to thank the Reviewer for this opinion.

  1. However, specific improvements should be considered regarding the general language, methodology, further controls regarding comparing the results with tradition methods such PCR or serology the should be considered.

Answer: We have made every effort to improve the quality of the English language. We agree with the Reviewer that further studies should be performed using other glycoprofile analysis techniques. This information is included in the summary section.

  1. Abstract and conclusion may also need to be improved, as the objective and aim are not included in the abstract, while further recommendations should be included in the conclusion.

Answer: The changes and supplementations have been made in the appropriate places (see the Abstract and Conclusions).

  1. I think references are appropriate and updated.

Answer: Thank you for this opinion.

  1. I think the data of the results are well presented in tables and figures with satisfied quality.

Answer: We would like to thank the Reviewer for this comment.

Comments on the Quality of English Language: Needs a little bit improvement

Answer: We would like to thank the Reviewer for his opinion. As it was mentioned above (point 4), we have made every effort to improve the quality of the English language.